# Impact of COPD Treatment on Survival in Patients with Advanced Non-Small Cell Lung Cancer

**DOI:** 10.3390/jcm11092391

**Published:** 2022-04-24

**Authors:** Hyunji Jo, Sojung Park, Nam Eun Kim, So Young Park, Yon Ju Ryu, Jung Hyun Chang, Jin Hwa Lee

**Affiliations:** Department of Internal Medicine, College of Medicine, Ewha Womans University, Seoul 07804, Korea; hyunji83085@gmail.com (H.J.); bsboys78@gmail.com (S.P.); gnikcor88@gmail.com (N.E.K.); sy.park12@gmail.com (S.Y.P.); medyon@ewha.ac.kr (Y.J.R.); hs1017@ewha.ac.kr (J.H.C.)

**Keywords:** bronchodilator, chronic obstructive pulmonary disease, inhaled corticosteroids, non-small cell lung cancer, survival

## Abstract

Chronic obstructive pulmonary disease (COPD) is associated with a poor prognosis in patients with non-small cell lung cancer (NSCLC). However, the impact of COPD treatment on the survival of patients with advanced NSCLC remains uncertain. We retrospectively investigated COPD patients among patients newly diagnosed with advanced NSCLC between September 2005 and August 2019 at a university hospital. The clinical characteristics, lung function, and survival outcomes were analyzed and compared between patients who did and did not receive COPD treatment. Among 221 patients with advanced NSCLC and COPD, 124 patients received treatment for COPD and 97 patients did not receive treatment for COPD. Forced expiratory volume in 1 s (FEV_1_) % predicted value was greater in the no-treatment group than in the COPD treatment group (*p* < 0.001). The median overall survival (OS) of the treatment group was 10.7 months, while that of the no-treatment group was 8.7 months (*p* = 0.007). In the multivariate analysis, COPD treatment was independently associated with improved OS (hazard ratio 0.71, 95% confidence interval 0.53–0.95, and *p* = 0.021). COPD treatment was associated with improved OS in patients with advanced NSCLC and COPD. Therefore, pretreatment spirometry and maximal treatment for COPD may offer a chance of optimal management for patients with advanced NSCLC.

## 1. Introduction

Lung cancer is the leading cause of cancer-related deaths worldwide. Chronic obstructive pulmonary disease (COPD) is an important comorbidity of lung cancer, sharing a common risk factor, i.e., cigarette smoking. Around 30–70% of lung cancer patients also have COPD [1,2,3]. COPD is associated with an increased risk during surgical resection, which offers the best prospect of long-term survival in patients with anatomically resectable non-small cell lung cancer (NSCLC) [4,5,6]. Even in the operable stage of NSCLC, patients with severe COPD generally cannot undergo surgery due to their high risk. Therefore, for patients with resectable NSCLC, spirometry is an essential test before surgery. Older smokers who have not experienced dyspnea, probably due to their habitual low activity, are often diagnosed with COPD for the first time as a result of spirometry during a pre-surgical examination for NSCLC. Several studies have reported that preoperative treatment with long-acting muscarinic antagonist (LAMA), long-acting β_2_-agonist (LABA), and/or inhaled corticosteroids (ICS), reduces the incidence of postoperative complications [7,8,9,10].

Similarly, in advanced NSCLC, several studies have reported that forced expiratory volume in 1 s (FEV_1_) less than 50% predicted or the presence of COPD were associated with mortality [11,12,13]. Because severity of airflow obstruction and acute exacerbations are predictors of mortality in COPD patients [14,15,16], improving these factors might also be important in patients with advanced NSCLC and COPD. Particularly, ICS has been associated with decreased mortality in COPD [17,18,19]. Nevertheless, few studies have investigated whether COPD treatment improves the survival of patients with advanced NSCLC.

The aim of the present study was to investigate the impact of COPD treatment on survival in patients with advanced NSCLC.

## 2. Materials and Methods

### 2.1. Study Population

We retrospectively investigated patients who were newly diagnosed with advanced NSCLC (inoperable stage IIIA to IV) between September 2005 and August 2019 at the Ewha Womans University Mokdong Hospital, Seoul, Korea (Figure 1). Among these, patients with spirometrically diagnosed COPD or patients who had already received treatment for COPD were included in the present study. The post-bronchodilator FEV_1_/forced vital capacity (FVC) < 0.7 was applied to define COPD. Patients were excluded if they had undergone surgical resection, did not have a histological diagnosis, were transferred to another hospital, or dropped out immediately after the diagnosis. We also excluded patients who harbored target mutations, such as epidermal growth factor receptor mutation and anaplastic lymphoma kinase rearrangement, and consequently received targeted therapies since they had a much better prognosis [20].

### 2.2. Data Collection and Assessment

Baseline clinical characteristics including age, sex, body mass index (BMI), smoking history, histological type, lung cancer stage, cancer treatment-related matters, and COPD treatment regimen were collected retrospectively using medical records. To diagnose COPD and determine the severity of airflow obstruction, spirometry performed within one year before and after the diagnosis of lung cancer was used.

The study subjects were divided into two groups according to whether they received COPD treatment or not. The COPD treatment group included patients who received inhaled therapy and/or theophylline. The no-treatment group included those who did not receive any treatment for COPD. The primary endpoint of the present study was to compare overall survival (OS) among these groups. In addition, we also investigated whether there was a difference in OS according to the COPD treatment regimens. OS was defined as the time from histologic diagnosis of NSCLC to death.

### 2.3. Statistical Analysis

The Pearson chi-square test or Fisher’s exact test was used to compare categorical variables, and the Student’s *t*-test was used to compare continuous variables. OS curves were plotted using the Kaplan–Meier method and were compared using a log-rank test. Hazard ratios (HRs) for univariate and multivariate survival analyses were calculated using the Cox proportional hazard model. All tests of significance were 2-sided, and differences among groups were considered significant when *p* < 0.05. All statistical analyses were performed using SPSS for Windows, version 27.0 (SPSS Inc., Chicago, IL, USA).

## 3. Results

### 3.1. Baseline Characteristics

A total of 221 patients with coexisting advanced NSCLC and COPD were included in the present study. Among these, 124 patients received COPD treatment and 97 patients did not receive COPD treatment. The baseline demographic and clinical characteristics of the patients are presented in Table 1. The mean age of the 221 patients was 70.7 ± 9.0 years old and 200 (90.5%) patients were men. There was a significant difference in lung cancer stage, where 61.9% of the no-treatment group had stage IV disease, while only 44.4% of the COPD treatment group had stage IV disease (*p* = 0.010). A total of 190 (86.0%) patients were newly diagnosed with COPD at the time of diagnosis of advanced NSCLC. A significantly higher proportion of patients in the no-treatment group were newly diagnosed COPD patients as compared with the COPD treatment group (99.0% vs. 75.8%, *p* < 0.001). In addition, the no-treatment group showed better baseline FVC, FEV_1_, and carbon monoxide diffusing capacity as compared with the COPD treatment group (Table 2). One hundred and five (84.7%) patients received inhalation therapy; triple therapy (ICS/LABA/LAMA) was the most commonly used regimen (45/124, 36.3%), followed by LAMA/LABA combination (22/124, 17.7%) and LAMA alone (20/124, 16.1%, Table 3).

### 3.2. Overall Survival

The treatment group showed significantly improved median OS as compared with the no-treatment group (10.7 months, 95% confidence interval (CI) 8.3–13.0, vs. 8.7 months, 95% CI 7.5–9.9, *p* = 0.007, Figure 2A). We also investigated whether there was a difference in OS according to treatment regimens. The Kaplan–Meier survival curves revealed that all types of COPD treatment were associated with improved median OS. Patients who received inhalation therapy showed a median OS of 10.9 months, while the other patients had a median OS of 8.9 months (*p* = 0.021, Figure 2B). The use of ICS (12.2 months vs. 8.8 months, *p* = 0.006, Figure 2C) and theophylline (11.4 months vs. 8.7 months, *p* = 0.016, Figure 2D) also resulted in significantly improved OS versus the untreated patients.

In subgroup analysis according to clinical stages, the median OS was significantly different according to COPD treatment status in stage III (12.0 months vs. 10.4 months, *p* = 0.032, Appendix A); however, there was no significant difference in stage IV (7.2 months vs. 6.6 months, *p* = 0.366, Appendix A).

Comparing OS according to the Global Initiative for Chronic Obstructive Lung Disease (GOLD) classification, on the one hand, the median OS of the COPD treatment group was significantly longer than that of the no-treatment group in GOLD 2 (11.8 months vs. 8.9 months, *p* = 0.007, Table 2). On the other hand, there was no significant difference in OS according to the presence or absence of COPD treatment in GOLD 1, GOLD 3, and GOLD 4 (Table 2).

In univariate analysis, sex, BMI, FEV_1_ less than 50% predicted, FVC, stage, chemotherapy, COPD treatment, inhaler use, ICS use, and theophylline-containing treatment were revealed as significant prognostic factors predicting OS (all *p* < 0.05). In the multivariate Cox regression analysis, men (HR 2.49, 95% CI 1.52–4.10, and *p* < 0.001) and stage IV disease (HR 1.94, 95% CI 1.44–2.62, and *p* < 0.001) were independent predictors of a worse OS, whereas higher BMI (HR 0.95, 95% CI 0.91–0.99, and *p* = 0.033), chemotherapy (HR 0.44, 95% CI 0.31–0.62, and *p* < 0.001), and COPD treatment (HR 0.71, 95% CI 0.53–0.95, and *p* = 0.021) were independent predictors of better OS (Table 4). When applying inhalation therapy or ICS instead of COPD treatment with the same variables, inhalation therapy (HR 0.72, 95% CI 0.54–0.97, and *p* = 0.029; Appendix A) and ICS were independent predictors of a better OS (HR 0.67, 95% CI 0.49–0.92, and *p* = 0.012; Appendix A). However, theophylline administration was not an independent predictor of OS (HR 0.79, 95% CI 0.58–1.08, and *p* = 0.135; Appendix A).

## 4. Discussion

The current study demonstrated that COPD treatment was associated with improved OS in patients with advanced NSCLC, even though the COPD treatment group had worse baseline lung function than the no-treatment group. When OS was analyzed for each type of COPD treatment, inhalation therapy and the use of ICS were associated with improvements in OS.

COPD is an important comorbidity for lung cancer patients. Therefore, several studies have investigated the relationship between COPD and the prognosis of lung cancer patients. In patients with inoperable NSCLC, FEV_1_ < 60% predicted or diffusing lung capacity < 60% predicted have both been associated with a worse prognosis [11,21]. FEV_1_ itself is a predictor of mortality, not only in COPD patients but also in the general population [14,22]. For patients with operable lung cancer, preoperative spirometry is essential, whereas spirometry may not be necessary for patients with inoperable lung cancer, particularly if they do not complain of respiratory symptoms, in which case clinicians are likely to focus only on lung cancer. Since physical activity usually decreases with age, patients with mild to moderate COPD, usually defined as FEV_1_ ≥ 50% predicted, do not experience or overlook their respiratory symptoms in many cases. In this study, 86% of patients with advanced NSCLC were diagnosed with COPD for the first time. Unfortunately, out of 190 patients with newly diagnosed COPD, only 94 (49%) patients were treated for their COPD. These results indicate that the identification of COPD through spirometry and active treatment for COPD is essential before treatment of their cancer.

As a result of comparing the OS with and without COPD treatment according to the GOLD classification, the OS of the COPD treatment group was significantly improved as compared with that of the no-treatment group in GOLD 2. In GOLD 1, GOLD 3, and GOLD 4, there was no difference in OS according to the presence or absence of COPD treatment. Since more than half of our study subjects belonged to the GOLD 2 stage (121/221) and the number of GOLD 1, GOLD 3, and GOLD 4 patients was relatively small, it is most likely that only GOLD 2 stage showed OS differences depending on the presence or absence of COPD treatment. However, similar to our study results, a study examining the effects of salmeterol/fluticasone propionate (SFC) according to the GOLD stage also showed that SFC significantly reduced mortality as compared with the placebo in GOLD 2 only [23].

In COPD patients, inhaled therapy, including LAMA, LABA, ICS, and combination therapy, have shown improvements in lung function, quality of life, and exacerbations [18,24,25,26]. However, the GOLD noted a lack of convincing evidence for a survival benefit with inhaled therapy in COPD. Recently, large randomized trials have shown a survival benefit of ICS-containing combination therapy over dual bronchodilator therapy with LAMA-LABA [18,19]. Interestingly, according to a nationwide study of COPD patients in Korea, ICS users appear to have a lower risk of lung cancer as compared with nonusers [27]. It is hypothesized that ICS, which reduces the risk of acute exacerbations through anti-inflammatory action in COPD patients, reduces the risk of lung cancer, a type of chronic inflammation. In this study, ICS was shown to prolong the survival of patients with advanced NSCLC. It is possible that the anti-inflammatory action of ICS inhibits cancer progression. There is also a report that ICS lowered the risk of coronary heart disease in COPD patients [28]. Therefore, the use of ICS may have had an effect on cardiovascular mortality in our study subjects. Since detailed information on the cause of death was not available for many of our subjects, it was not possible to evaluate whether inhaled therapy was associated with cancer progression or other fatal complications, such as exacerbation of COPD, pneumonia, and cardiovascular events. Inhaled therapy may have also influenced their performance status, which is a major factor in determining whether to treat lung cancer, and is itself an important prognostic factor in lung cancer patients [29].

In COPD patients, ICS increases the risk of pneumonia [30,31,32]. Pneumonia is one of the most life-threatening complications of lung cancer patients. However, previous studies have demonstrated consistent results that the use of ICS does not increase pneumonia-related mortality in COPD patients, and instead it is associated with decreased mortality in hospitalized pneumonia patients [17,33]. This might suggest a double effect of ICS, i.e., an immunosuppressive effect and an anti-inflammatory effect. ICS, which achieves locally high concentrations in the lung, upregulates the production of anti-inflammatory proteins and inhibits the transcription of proinflammatory cytokines and chemokines [34]. In a previous study of patients receiving cisplatin-containing chemotherapy, inhaled fluticasone reduced the incidence of delayed pulmonary toxicity [35]. Because there are also reports that different types of ICS have different impacts on the incidence of pneumonia, we need to make careful choices about ICS type and dose [31,36].

The present study has several limitations. First, it was a single-center study and not all patients with newly diagnosed advanced NSCLC had performed spirometry, therefore, there may have been selection bias. Second, it was not possible to investigate their performance status. Performance status is an important prognostic factor and may affect treatment decision making [37,38,39]. Nevertheless, BMI and undergoing cytotoxic chemotherapy might indirectly reflect their performance status, and these factors were adjusted for the analysis. Third, it is not clear why the no treatment group did not receive COPD treatment. As this study was a retrospective study, the respiratory symptoms of the study subjects were not evaluated. Nevertheless, the fact that the no treatment group had better lung function than the COPD treatment group strongly suggests the possibility of not receiving COPD treatment due to no respiratory symptoms. Even if a doctor tried to prescribe an inhalation drug, it is likely that a patient with no symptoms refused it. Fourth, although COPD is characterized by not fully reversible airflow obstruction, we were unable to evaluate whether lung function or dyspnea improved after treatment in the COPD treatment group. Fifth, although there was no statistically significant difference in the type of first-line chemotherapy between the two groups, the possibility that immune checkpoint inhibitors might affect the survival should be considered. In this study, four patients of the COPD treatment group received immune checkpoint inhibitors. Immune checkpoint inhibitors were administered as a second-line treatment to one patient and as a later line of treatment to three patients. These patients received two to four cycles of immune checkpoint inhibitors; however, there was no treatment response.

## 5. Conclusions

COPD treatment is associated with improved survival in patients with both advanced NSCLC and COPD. Additionally, this study showed that ICS and inhalation therapy are associated with improved OS. In patients with advanced NSCLC, finding COPD through spirometry before cancer treatment and treating the COPD as well as the lung cancer could improve survival.

## Figures and Tables

**Figure 1 jcm-11-02391-f001:**
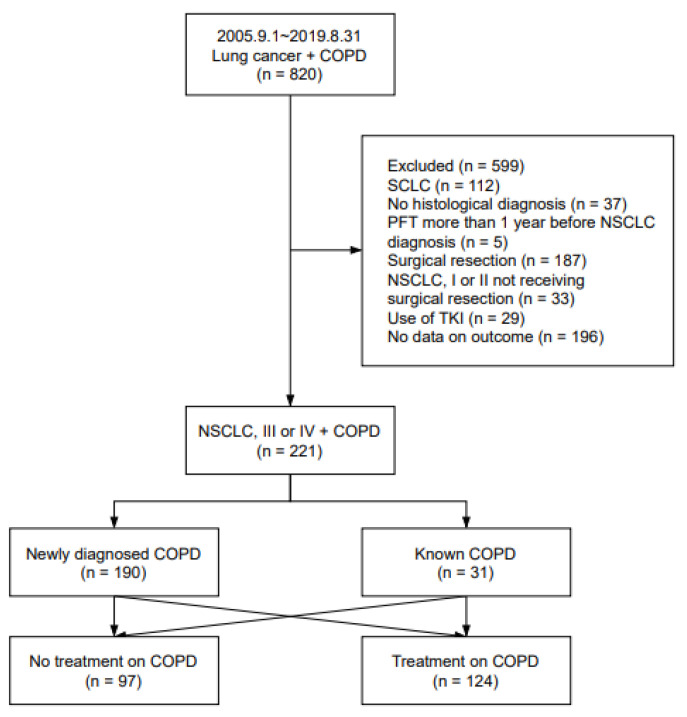
Study design. COPD, chronic obstructive pulmonary disease; SCLC, small cell lung cancer; PFT, pulmonary function test; NSCLC, non-small cell lung cancer; TKI, tyrosine kinase inhibitor.

**Figure 2 jcm-11-02391-f002:**
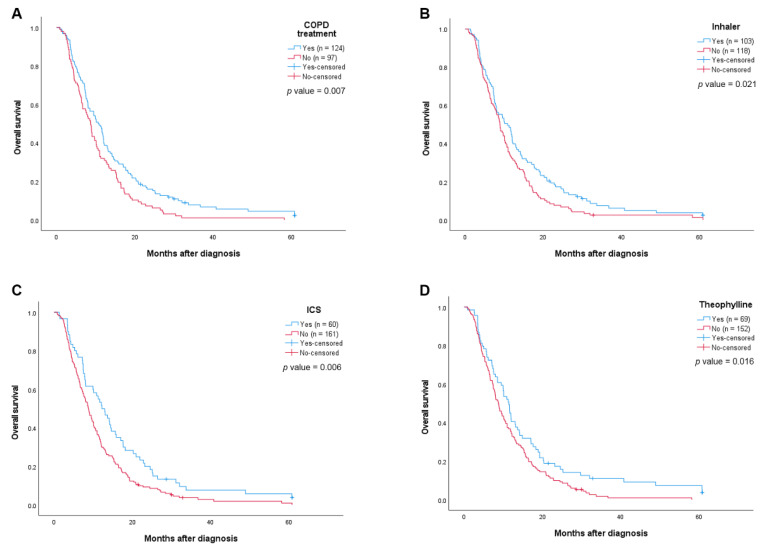
Kaplan–Meier curves of overall survival of advanced NCSLC: (**A**) Overall survival curves by COPD treatment; (**B**) overall survival curves by use of inhaler; (**C**) overall survival curves by use of ICS; (**D**) overall survival curves by use of theophylline.

**Table 1 jcm-11-02391-t001:** Comparison of baseline characteristics by COPD treatment status.

	Total *n* = 221	Treatment *n* = 124	No Treatment *n* = 97	*p*-Value
Sex, men	200 (90.5)	114 (91.9)	86 (88.7)	0.410
Age, years	70.7 ± 8.97	71.2 ± 7.79	70.0 ± 10.3	0.330
BMI, kg/m^2^	22.3 ± 3.19	22.3 ± 3.29	22.4 ± 3.06	0.873
Smoking				0.084
Never smoker	37 (16.7)	16 (12.9)	21 (21.6)	
Ever smoker	184 (83.3)	108 (87.1)	76 (78.4)	
Histology				0.090
SqCC	117 (52.9)	70 (56.5)	47 (48.5)	
ADC	77 (34.8)	36 (29.0)	41 (42.3)	
P/D carcinoma	27 (12.2)	18 (14.5)	9 (9.3)	
COPD diagnosis				<0.001
New COPD	190 (86.0)	94 (75.8)	96 (99.0)	
Known COPD	31 (14.0)	30 (24.2)	1 (1.0)	
Clinical stage				0.010
III	106 (48.0)	69 (55.6)	37 (38.1)	
IV	115 (52.0)	55 (44.4)	60 (61.9)	
Chemotherapy				0.451
Chemotherapy	165 (74.7)	95 (76.6)	70 (72.2)	
No chemotherapy	56 (25.3)	29 (23.4)	27 (27.8)	
First-line chemotherapy				0.098
Platinum + gemcitabine	48 (29.1)	34 (35.8)	14 (20.0)	
Platinum + pemetrexed	32 (19.4)	16 (16.8)	16 (22.9)	
Platinum + taxane	56 (33.9)	26 (27.4)	30 (42.9)	
Taxane only	17 (10.3)	11 (11.6)	6 (8.6)	
Others	12 (7.3)	8 (8.4)	4 (5.7)	
Radiotherapy				0.452
CCRT	54 (24.4)	33 (26.6)	21 (21.6)	
Palliative	12 (5.4)	5 (4.0)	7 (7.2)	
No radiotherapy	155 (70.1)	86 (69.4)	69 (71.1)	
Follow up duration, months	9.5 (5.3–16.1)	10.7 (5.8–18.3)	8.7 (4.4–15.1)	0.013

Data are shown as *n* (%) per each group or means ± standard deviation or median (interquartile range). COPD, chronic obstructive pulmonary disease; BMI, body mass index; SqCC, squamous cell carcinoma; ADC, adenocarcinoma; P/D, poorly differentiated; CCRT, concurrent chemoradiation therapy.

**Table 2 jcm-11-02391-t002:** Comparison of lung function by COPD treatment status.

	Total *n* = 221	Treatment *n* = 124	No Treatment *n* = 97	*p*-Value
FVC, liters	2.91 ± 0.81	2.81 ± 0.76	3.04 ± 0.86	0.035
FVC % predicted	77.8 ± 18.2	75.7 ± 17.9	80.5 ± 18.3	0.053
FEV_1_, liters	1.68 ± 0.55	1.53 ± 0.48	1.87 ± 0.58	<0.001
FEV_1_ % predicted	65.3 ± 18.7	60.2 ± 16.8	71.8 ± 19.0	<0.001
FEV1/FVC	57.9 ± 10.4	55.1 ± 11.2	61.5 ± 7.90	<0.001
DLco % predicted	68.7 ± 27.5 ^a^	64.7 ± 25.8 ^b^	74.0 ± 28.8 ^c^	0.033
COPD severity by GOLD classification ^d^				<0.001
GOLD 1	52 (23.5)	17 (13.7)	35 (36.1)	
GOLD 2	121 (54.8)	72 (58.1)	49 (50.5)	
GOLD 3	44 (19.9)	33 (26.6)	11 (11.3)	
GOLD 4	4 (1.8)	2 (1.6)	2 (2.1)	
Median overall survival by GOLD classification ^e, f^				
GOLD 1	9.7 (7.3–12.1)	11.7 (6.3–17.0)	9.0 (6.9–11.1)	0.414
GOLD 2	10.6 (8.4–12.9)	11.8 (9.8–13.8)	8.9 (6.6–11.2)	0.007
GOLD 3	7.4 (5.2–9.6)	7.9 (4.9–10.9)	6.4 (5.2–9.6)	0.253
GOLD 4	3.9 (0.8–7.0)	4.0 (NE)	0.8 (NE)	0.090

Data are shown as *n* (%) per each group or means ± standard deviation. ^a^ Data are from 160 patients. ^b^ Data are from 90 patients. ^c^ Data are from 70 patients. ^d^
*p*-Value was calculated from the Fisher’s exact test. ^e^ Data are shown as months (95% confidence interval). ^f^
*p*-Value was calculated from log-rank test. COPD, chronic obstructive pulmonary disease; FVC, forced vital capacity; FEV_1_, forced expiratory volume in 1 s; DLco, diffusing capacity of carbon monoxide; GOLD, the Global Initiative for Chronic Obstructive Lung Disease; NE, not estimated. GOLD 1, FEV_1_ ≥ 80% predicted; GOLD 2, 50% ≤ FEV_1_ < 80% predicted; GOLD 3, 30% ≤ FEV_1_ < 50% predicted; GOLD 4, FEV_1_ < 30% predicted.

**Table 3 jcm-11-02391-t003:** Types of COPD treatment.

Type of Treatment	*n* = 124
Inhalers	
LAMA	20 (16.1)
LABA	2 (1.6)
LAMA/LABA	22 (17.7)
ICS/LABA	16 (12.9)
ICS/LAMA/LABA	45 (36.3)
No use	19 (15.3)
Theophylline	
Use	69 (55.6)
No use	55 (44.4)

Data are presented as number (%). LAMA, long-acting muscarinic antagonist; LABA, long-acting β_2_-agonist; ICS, inhaled corticosteroids.

**Table 4 jcm-11-02391-t004:** Clinical factors associated with overall survival.

Variables	Univariate			Multivariate		
	HR	95% CI	*p*-Value	HR	95% CI	*p*-Value
Sex, men	1.62	1.02–2.56	0.042	2.49	1.52–4.10	<0.001
Age, years	1.00	0.98–1.02	0.846	0.99	0.98–1.01	0.313
BMI, kg/m^2^	0.95	0.91–0.99	0.038	0.95	0.91–0.99	0.033
Ever smoker	1.02	0.72–1.46	0.896			
FEV_1_ < 50% predicted	1.40	1.01–1.95	0.044	1.26	0.90–1.79	0.184
Histology						
ADC	1.00					
SqCC	1.05	0.79–1.41	0.736			
P/D carcinoma	1.17	0.75–1.81	0.497			
Clinical stage						
III	1.00			1.00		
IV	1.62	1.24–2.13	<0.001	1.94	1.44–2.62	<0.001
Chemotherapy	0.59	0.44–0.81	0.001	0.44	0.31–0.62	<0.001
CCRT	0.98	0.66–1.24	0.542			
COPD treatment	0.69	0.52–0.91	0.007	0.71	0.53–0.95	0.021
Inhaled therapy	0.73	0.55–0.95	0.021			
ICS	0.65	0.48–0.89	0.006			
Theophylline	0.70	0.52–0.94	0.017			

HR, hazard ratio; CI, confidence interval; BMI, body mass index; FEV_1_, forced expiratory volume in 1 s; ADC, adenocarcinoma; SqCC, squamous cell carcinoma; P/D, poorly differentiated; CCRT, concurrent chemoradiation therapy; COPD, chronic obstructive pulmonary disease; ICS, inhaled corticosteroids.

## Data Availability

The datasets used and analyzed during the current study are available from the corresponding author on reasonable request.

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
