# Peer review of "Impact of COPD Treatment on Survival in Patients with Advanced Non-Small Cell Lung Cancer"

_jcm, 2022, doi:10.3390/jcm11092391_

Round 1

Reviewer 1 Report

The authors of the manuscript, entitled “Impact of COPD treatment on survival in patients with advanced non-small cell lung cancer” evaluated COPD treatment effect on general cancer treatment outcome for patients with advanced non-small cell lung cancer. The study appears to be well-conducted, however, it has several limitations which authors discussed and my comments are outlined below.  

  1. As authors discussed, the main concern is the potential other confounding factors might have been impacted on the treatment outcome, such as performance status, lung cancer disease burden or therapeutic modalities etc. Authors indicated the number of patients with stage IV disease for each group, however, not every stage IV is same with regard to disease burden and authors need to clarify if any of patients received checkpoint blockade immunotherapy which has been used since 2015.
  2. Authors stratified patients based on COPD GOLD stage in each group, however, the survival was not analyzed based on GOLD stage. The stage I COPD patients were more in the non-treatment group, yet, the survival was worse compared to treated group. So, survival analysis based on GOLD stage my provide additional insight regarding COPD treatment effect.
  3. Figure 2 survival curve is very blurry, I am not able to see the numbers on the figures.

Author Response

Point 1: As authors discussed, the main concern is the potential other confounding factors might have been impacted on the treatment outcome, such as performance status, lung cancer disease burden or therapeutic modalities etc. Authors indicated the number of patients with stage IV disease for each group, however, not every stage IV is same with regard to disease burden and authors need to clarify if any of patients received checkpoint blockade immunotherapy which has been used since 2015.

Response 1: Thank you for your comments.

Your points are important. In this cohort, 4 patients of COPD treatment group received checkpoint blockade immunotherapy; 2 cycles of atezolizumab as second-line, 3 cycles of atezolizumab as third-line, 2 cycles of atezolizumab as fouth-line, and 4 cycles of pembrolizumab as fouth-line. However, given the progression-free survival of immunotherapy in previously treated advanced NSCLC, we speculated that immunotherapy might not affect survival because the duration of treatment was too short. Rather than the possibility that immunotherapy improved OS, it is possible that COPD treatment improved the performance status of patients with advanced NSCLC, providing an opportunity for immunotherapy that can only be administered as a 2nd line or later in Korea.

In response to your comments, we have added the following sentences to the discussion section.

‘Fifth, although there was no statistically significant difference in the type of first-line chemotherapy between the two groups, the possibility that immune checkpoint inhib-itors might affect the survival should be considered. In this study, 4 patients of the COPD treatment group received immune checkpoint inhibitors. Immune checkpoint inhibitors were administered as a second-line treatment to one patient and as a later line of treatment to three patients. These patients received 2 to 4 cycles of immune checkpoint inhibitors; however, there was no treatment response.’

We agree with your point that not every stage IV is the same with regard to disease burden. Although the burden of disease in stage IV patients in this study could not be investigated in detail, as shown in Figure S1B of Additional file 1, there was no difference in OS with or without COPD treatment for stage IV patients only.

Point 2: Authors stratified patients based on COPD GOLD stage in each group, however, the survival was not analyzed based on GOLD stage. The stage I COPD patients were more in the non-treatment group, yet, the survival was worse compared to treated group. So, survival analysis based on GOLD stage my provide additional insight regarding COPD treatment effect.

Response 2: Thank you for your comments. We agree on your opinion.

According to your comments, we have added survival analysis based on GOLD stage into Table 2. Also we have added the following senteces to the results and the discussion section.

“Comparing the OS according to the Global Initiative for Chronic Obstructive Lung Disease (GOLD) classification, the median OS of the COPD treatment group was significantly longer than that of the no-treatment group in GOLD 2 (11.8 months vs. 8.9 months, P = 0.007, Table 2). On the other hand, there was no significant difference in OS according to the presence or absence of COPD treatment in GOLD 1, GOLD 3, and GOLD 4 (Table 2).”

“As a result of comparing the OS with and without COPD treatment according to the GOLD classification, the OS of the COPD treatment group was significantly improved compared with that of the no-treatment group in GOLD 2. In GOLD 1, GOLD 3, and GOLD 4, there was no difference in OS according to the presence or absence of COPD treatment. Since more than half of our study subjects belonged to GOLD 2 stage (121/221) and the number of GOLD 1, GOLD 3, and GOLD 4 patients was relatively small, it is most likely that only GOLD 2 stage showed OS differences depending on the presence or absence of COPD treatment. However, similar to our study results, a study examining the effects of salmeterol/fluticasone propionate (SFC) according to the GOLD stage also showed that SFC significantly reduced mortality compared to placebo in GOLD 2 only [23].”

Point 3: Figure 2 survival curve is very blurry, I am not able to see the numbers on the figures.

Response 3: We apologize for the inconvenience. We modified the size of Figure 2.

Reference

  1. Jenkins C.R., Jones P.W., Calverley P.M., Celli B., Anderson J.A., Ferguson G.T., Yates J.C., Willits L.R., Vestbo J. Efficacy of salmeterol/fluticasone propionate by GOLD stage of chronic obstructive pulmonary disease: Analysis from the randomised, placebo-controlled TORCH study. Respir. Res. 2009;10:59. doi: 10.1186/1465-9921-10-59.

Reviewer 2 Report

This research has certain innovation and clinical practice significance.

It is recommended that different regimens of chemotherapy be included in the analysis.

Author Response

Point 1: This research has certain innovation and clinical practice significance.

It is recommended that different regimens of chemotherapy be included in the analysis.

Response 1: Thank you for your comment.

Based on your comment, we have added first-line chemotherapy of included patients to the Table 1. As a result, there was no statistically significant difference in the type of chemotherapy between the two groups.

Round 2

Reviewer 1 Report

Authors responded to my concerns and comments satisfactorily. I still see the figure is blurry, otherwise, no other concern. 

Reviewer 2 Report

This article responds to the review comments one by one